# The Role of Information Systems in the Sustainable Development of Enterprises: A Systematic Literature Network Analysis

**Fan Zeng [1], Stacy Hyun Nam Lee [2] and Chris Kwan Yu Lo [1,*]**

[1] Business Division, Institute of Textiles and Clothing, Hong Kong Polytechnic University, Kowloon, Hong Kong 999077, China; fan.zeng@connect.polyu.hk

[2] Department of Hospitality and Retail Management, Texas Tech University, Lubbock, TX 79409, USA; stacy.h.lee@ttu.edu

[*] Correspondence: kwan.yu.lo@polyu.edu.hk; Tel.: +852-2766-5609

**Abstract:** Information Systems (IS) play an important role in improving the efficiency of firms' operations and supply chains, which links to sustainability. Therefore, this study conducted a systematic literature network analysis to review 132 articles that discuss current trends in the IS discipline. Based on a citation network analysis, this study discovered three main research domains (sustainable competitive advantage, environmental sustainability, and sustainable online social communities), and two emerging research domains (the role of IS in developing countries and sustainable information infrastructures). Furthermore, a main path analysis was conducted to understand the knowledge structure of each research domain. This addresses how different trends are reflected in the IS literature related to improving firms' competitive advantages and environmental sustainability. The results found that the sustainable competitiveness of enterprises is improved by the synergy between IS and other recourses within in the enterprises. Green IS initiatives not only solve the issues of environmental sustainability, but also enhance sustainable competitive advantage (i.e., stock price). As social media becomes the optimal enterprise communication channel, this study discusses the factors affecting sustainable online social community, such as structural dynamics (i.e., membership size, communication activity), social dynamics (the basic nature of interactions among members), participation costs, and topic consistency. Overall, the Information System literature is highly focused on three areas—economy, environment, and society, which supports Triple Bottom Line theory.

**Keywords:** information systems; citation network analysis; sustainable competitive advantage; environmental sustainability; online community; sustainability; social media

---

## 1. Introduction

Information Systems (IS) put emphasis on "integrating information technology solutions and business processes to meet the information needs of businesses and other enterprises" (The Association for Computing Machinery, 2005, p.14). IS help to support enterprises' operations, management, and decision-making processes [1]. IS are not only comprised of the combination of information and communication technology (ICT), but also the way in which people interact with these technologies in support of business processes [2]. In this sense, well-functioning IS are special types of work systems in which humans or machines perform processes and activities that use resources to produce specific products or services for customers [3,4]. IS for enterprises' operations management is a pyramid of systems covering transaction processing systems at the bottom to management IS, decision support systems, and executive IS at the top, and supporting decision-making in every

part of the supply chain [5]. For example, IS are involved in each phase of the fast fashion supply chain, from fashion design (i.e., artificial neural network and smart retail systems), demand forecasting (i.e., big data, cloud computing, extreme learning machine), sourcing and production (i.e., RFID), inventory and transportation management (i.e., vendor managed inventory), to retailing (i.e., smart retail systems, return management systems) [6]. Therefore, enterprises implement digital transformation (i.e., information systems) so that they can maintain sustainable competitive advantages over other competitors.

As a source of sustainable competitive advantages, IS have helped improve enterprises' performances by increasing the effectiveness and efficiency of enterprises [7,8]. A sustainable competitive advantage is defined as "the long-term benefit of implementing some unique value-creating strategy which competitors do not implement simultaneously, along with the inability to duplicate the benefits of this strategy" ([9], p. 1612). IS can be a way of achieving sustainable competitive advantages by utilizing unique enterprise attributes [10]. Previous literature found that enterprises can improve firm performance by reducing costs and delivery time, and enhancing customer service and reliability with help of ICT implementation [11]. Consequently, the potential of IS to enhance management decision-making capabilities and productivity by adopting enterprise resource systems, supply chain collaboration support systems [12], and social reference systems to support sustainable social commerce sales, have been explored [13]. Although information systems play an important role in promoting the sustainable competitive advantage of the enterprise, there are still many problems in the implementation of an information system. A recent survey found that many executives are concerned about IS implementation because 70% of all IS initiatives did not reach their goals [14]. Out of USD $1.3 trillion that was spent on IT in 2018, approximately $900 billion was estimated as wasted [14]. The role of IS in the sustainable development of enterprises is an important issue that needs further study.

Moreover, due to the great demand of sustainability in consumer brand selection criteria nowadays [15,16], this has caused numerous companies to change their operations by eliminating usage of single-use plastics and using artificial intelligence to build greener production processes [17]. A number of enterprises have implemented IS to promote environmental and social sustainability [18,19]. For instance, JD.com implements information technology to establish the systems of electronic invoices, which helped to saved 91 tons of paper [20]. With the help of Augmented Reality (AR) technology, some retailers (i.e., ASOS, Amazon) give their consumer a picture of the clothes worn on their body in reality, which helps the consumer find suitable products and reduces carbon emissions from lower rate of refund [21]. Using enterprise resource planning application, supply chain management software, and Internet of Things technology, fashion brand Everlane provides transparent information for the whole production, which includes sourcing different components and costs, labor conditions, carbon footprint, and environmental implications [22]. In terms of IS academia, a high amount of literature has continuously made important contributions to research on sustainability also [23–27]. IS directly assist in environmental sustainability practices, such as implementing greenhouse gas emission tracking systems within a logistics process [28], an indexing system for green supplier evaluations [29], and a support system for the vehicle industry to improve the efficiency of electric vehicle battery usage [30].

Therefore, IS have helped enterprises to transform into environmentally sustainable entities to meet global environmental challenges and respond to consumers' requirements [19,23]. Due to the increasing attention towards the stainable development of enterprises in IS, this study aims to understand the role that IS play in the sustainable development of enterprises. To be specific, the objectives of this study are (1) to describe the nature of sustainability in IS related articles since 1993; (2) to identify the key research domains of sustainability issues in IS; and (3) to analyze how sustainability is discussed in IS literature and find key research gaps for future research. To achieve the objectives of this study, the Systematic Literature Network Analysis (SLNA) was used to understand the nature of sustainability in IS and key research domains, while the Main Path Analysis (MPA) was conducted to identify important milestones in the theoretical development.

## 2. Information Systems and Sustainable Development

Business value, from investment on IS, can only be generated through business improvement and innovation (i.e., product, service innovation, improved business process) [31]. In other words, sustainable competitive advantages should be realized through good cooperation between the operation of the enterprise and the IS [10]. To be specific, IS can be applied in operations management, including knowledge sharing in the entire supply chain, healthcare, and omni-channel retailing and recommendation systems [32]. In the past, a small number of IS review papers focused on the role of IS in the sustainable development of enterprises, such as Peppard and Ward [31], Wade and Hulland [33], and Piccoli and Ives [34]. Peppard and Ward [31] indicated that resource-based theory is suitable to explain the role of IS in sustainable competitive advantage. From the perspective of resource-based theory, the performance of an organization depends largely on IS capability [31]. To be specific, IS capability supports and improves the operations of enterprises, enhancing organizational performance in the long-term [31]. Wade and Hulland [33] also agreed that the complementarity of IS and other resources affected sustainable competitive advantage and highlighted that this is moderated by organizational factors (i.e., strong top managers) and environmental factors (i.e., stable business environment, turbulent business environment). To be specific, in the context of strategic initiatives, a sustainable competitive advantage is achieved by IT resource barriers (i.e., IT assets, IT capabilities), complementary resource barriers, IT project barriers (technology characteristics, implementation process), and preemption barriers (switching costs, value system structural characteristics).

The deterioration of the natural environment brings risks and opportunities to enterprises [35]. IS research can contribute to the knowledge link between information, organization, and natural environments, to the innovation of environmental strategy, to the creation and evaluation of eco-friendly systems, and to the improvement of the environment [35]. Therefore, Melville's [35] review paper discussed the factors that promote or inhibit the adoption of environmentally sustainable business practices (i.e., culture), the relationship between environmentally sustainable business practices and business performance, and the relationship between IS and the environmental performance of a supply chain. Enterprises are considered key contributors to environmental sustainability because of their global, national, and/or local innovation and rapid change capabilities [19]. Elliot [19] analyzed the challenges of environmental sustainability, including accessing the state of environmental deterioration, the acceleration towards deterioration caused by human activities, and the uncertainty of the human response to deterioration. They describe how people face these challenges through reviewing previous studies in the areas of environment, society, governments, industries and alliances, organizations, and individuals and groups within organizations.

In a word, previous articles only focused on one aspect. For example, Wade and Hulland [33] and Piccoli and Ives [34] focused on economic sustainability, while Melville [35] and Elliot [19] focused on environmental sustainability. However, according to Elkington's [36] Triple Bottom Line theory, economic sustainability, environmental sustainability, and social sustainability integrate in a whole organic unity. In addition, no review studies explore Systematic Literature Network Analysis (SLNA) in the description of the role of IS in the sustainable development of enterprises entirely and systematically. To fill the research gap, this study discusses the role of information systems in the sustainable development of enterprises from the perspective of integrating the economy, environment, and society.

## 3. Methodology

### 3.1. Study Design

In order to achieve the objectives of this study, a review study was employed to build a solid knowledge structure for IS and sustainability [37]. Consequently, this study employed SLNA to understand the role of IS in the sustainable development of enterprises, and then the findings were analyzed with MPA to identify the knowledge structure of research domains.

*Systematic literature network analysis:* Depending on the research objectives, different review types, such as review studies—narrative review, descriptive review, and theoretical review—can be considered. For the IS discipline, narrative reviews and theoretical reviews are the most common review studies [37]. Given that IT schemes are social-oriented technologies, IS require detailed systematic literature reviews, as these can point out relevant studies by using explicit methodology (i.e., main path analysis) [37,38]. Therefore, previous research [39,40] has also used SLNA to build up knowledge structures in specific research fields, such as supply chain risk management [41], occupational health and safety issues in operations management [39], and textile dyes and human health [40]. SLNA can integrate the advantages of systematic literature reviews and citation network analyses. Therefore, this review study employs SLNA, which combines SLR and CNA [41]. SLR refers to the assessment of literature "on a clearly formulated question that uses systematic and explicit methods to identify, select and critically appraise relevant primary research, and to extract and analyses data from the studies that are included in the rear-view" ([41], p.418). SLR can be beneficial in the understanding of current trends, to detect existing gaps in scientific literature, and to consolidate emerging topics in other areas [42]. In this sense, SLR can be used as a reliable technique to select appropriate keywords to locate relevant articles on a specific research topic, whereas CNA is suitable for analyzing the dynamics of knowledge evolution as well as building up the knowledge structure [41]. CNA is defined as "one form of social network in which authors and papers can be represented as nodes, and their mutual interactions (i.e., citations) can be modelled as edges." ([41], p. 418). CNA can help construct a path for the development of scientific thought [41]. The citation relationship is created between publications, which implies the spread of knowledge from one document to another, as literature focusing on the same research issues tend to cite each other based on previous knowledge [43,44]. Consequently, the software programs Gephi and Pajek were used to convert data for CNA, and further conduct the main path analysis to identify the significant publications in each research domain.

*Main path analysis:* To further examine the knowledge structure of the major research domains, an MPA was conducted through weighting the citations in the cluster to identify the most important citation path [39]. The main path analysis was conducted using Pajek 5.05, which helped choose the scheme of traversal weight and search for the main path. The traversal weight scheme included the Search Path Link Count (SPLC), Search Path Node Count (SPNC), and Search Path Count (SPLC) [43,45]. In term of the traversal weight scheme, SPLC was used, as it aids in the simulation of the situation of knowledge diffusion in scientific development, not only for conveying knowledge, but also for assigning the source of knowledge itself [46]. For searching in the main path, a global standard search was conducted, which provided the overall most significant main paths in the knowledge dissemination [41,47–49].

## 3.2. Data Collection

A keyword search was conducted using Web of Science (WOS) in June 2019 [50]. Different terms were used for IS in sustainability, such as "Green IS" [24]. The search terms used were "green" and "sustainable*" and these terms were searched for in titles, abstracts, keywords, and keyword plus [35,51,52]. More importantly, this study used the 8 journals indicated as the leading journals in IS areas by the Association for Information Systems [51]. A number of keywords were used as "TS = ("Green" OR "Sustainab*") AND SO = (*European Journal of Information Systems* OR *Information Systems Journal* OR *Information Systems Research* OR *Journal of the Association for Information Systems* OR *Journal of Information Technology* OR *Journal of Management Information Systems* OR *Journal of Strategic Information Systems* OR *MIS Quarterly*)." To be specific, only peer reviewed research and review articles from the period 1993 to 2018 were selected for this study. As a result, a total of 132 articles were collected for systematic literature network analysis. Detailed information is included in Table 1.

**Table 1.** The journals suggested by the Association for Information Systems.

| Name | Abbreviation |
| --- | --- |
| European Journal of Information Systems | EJIS |
| Information Systems Journal | ISJ |
| Information Systems Research | ISR |
| Journal of the Association of Information Systems | JAIS |
| Journal of Information Technology | JIT |
| Journal of Management Information Systems | JMIS |
| Journal of Strategic Information Systems | JSIS |
| MIS Quarterly | MISQ |

In order to identify key research domains, CNA was employed using a Girvan–Newman algorithm to categorize publications, construct research fields, and identify community structures of citation networks [39,53]. CNA allows for the visualization of research domains, which illustrates how studies are interrelated in a specific area and reveals how knowledge structures are created in specific domains [41,54,55]. In other words, the citation network presents a directed graph which serves as a network to embody a research domain [56]. In the directed graph, the nodes represent how articles are connected, and the links represent the citation relationship between articles [56]. To further understand knowledge structures, articles were sorted out into several research domains representing a topic in scientific literature [57]. Therefore, Girvan–Newman clustering was conducted using the network visualization software program, Gephi [58].

To find the community boundaries based on identical centrality indexes, Girvan and Newman [59] proposed an algorithm that could categorize communities in order to (1) calculate the betweenness for all edges in the network; (2) remove the edge with the highest betweenness; (3) recalculate the betweenness for all edges affected by the removal; and 4) repeat from Step 2 until no edges remain (p. 7823). They found that many networks present the attribute of community structure in which the network nodes are connected together in tightly knit groups, between which there are only looser connections. Therefore, each community in the citation network may exemplify publications that are related among specific topics [59]. Modularity refers to the measure of the quality of division of a network ([60], p. 8) In the symmetric matrix **e** (k × k), $e_{ij}$ the fraction of all edges in the network that link vertices in community **i** to vertices in community **j**. Therefore, the function of modularity is addressed as,

$$Q = \text{Tr}e - \| e^2 \| \tag{1}$$

where Tr$e$ represents the fraction of edges in the network that connect vertices in the same community, and $\| e^2 \|$ represents the sum of the elements $e_{ij}$ of the matrix **e** [60]. Its quantity measures the fraction of the edges in the network that connect vertices of the same type (i.e., within-community edges) minus the expected value of the same quantity in a network with the same community divisions, but with random connections between the vertices [60]. When the number of within-community edges, Q = 0, is no better than random, while Q = 1 is the maximum of modularity value, a strong community structure is indicated. Typically, values for such networks fall in the range of about 0.3 to 0.7 [60]. Therefore, Q ≈ 0.566 posits that the division of the network is effective. The modularity value reaches a peak value of 0.566 at 53 clusters (including the clusters that only have one paper), indicating that it is best to divide the citation network into 53 clusters (Q ≈ 0.566) [39,60].

To further examine the knowledge structure of the three major research domains, a main path analysis was conducted through weighting the citations in the cluster to identify the most important citation path [39]. The main path analysis was conducted using Pajek 5.05, which helped us to choose the scheme of traversal weight and search for the main path. The traversal weight scheme included the Search Path Link Count (SPLC), Search Path Node Count (SPNC), and Search Path Count (SPLC) [43,45]. In terms of the traversal weight scheme, SPLC was used, as it aids in simulating the situation of knowledge diffusion in scientific development, not only for conveying knowledge, but

also for assigning the source of knowledge itself [46]. For searching in the main path, a global standard search was conducted, which provided the overall most significant main paths in the knowledge dissemination [41,47–49]. The overall flow of data collection and data analysis is shown on Figure 1.

**Figure 1.** The flow of data collection and data analysis.

## 4. Findings

To understand the overview of the status of sustainability in IS literature, SLR was employed by collecting a total of 132 articles through a keyword search, and descriptive statistics were extracted. For descriptive statistics, several factors were examined, such as journal distribution, period of publication between 1993–2019, methodologies, and research context depending on industry and country. Subsequently, MPA was conducted to identify research domains in the theoretical development.

### 4.1. Results from SLR

#### 4.1.1. Distribution of Articles by Journal

Based on the Association for Information System criteria, 132 articles from the eight leading journals were analyzed (Appendix A). From the list of eight journals, articles that were related to sustainability and IS included 25% from MISQ, 19% from JSIS, 12% from EJIS, 12% from JMIS, 9% from JAIS, 8.33% from ISJ, 6% from JIT, and 4% from ISR (Figure 2).

#### 4.1.2. Distribution of Articles by Year of Publication

A total of 132 articles published from the period 1993 to 2018 were collected (Figure 3). Based on the yearly distribution of articles, the number of articles related to sustainability in IS gradually

increased until 2011. Although there was a slight reduction of articles from the period 2004 to 2008, there was an incremental escalation of sustainability in IS articles until 2018. However, the peak years for IS articles among the eight journals were found to be between 2011 and 2017. This may be because the United Nations set up 17 sustainable development goals in 2015, which triggered more research interest in sustainability [61]. In addition, news related to sustainability and environmental issues were highlighted in those years [62]. Consequently, many articles related to environmental sustainability have been published since 2009 [19,27,35]. MISQ published a special issue on IS and environmental sustainability in 2013. Furthermore, the Association for Information Systems established a special interest group for green research in 2010 to promote better solutions for preventing environmental degradation and climate change [23]. Based on the yearly distribution of articles, the average rate of IS articles published between the period 1993 and 2018 is reported to be 5.08 articles per year.

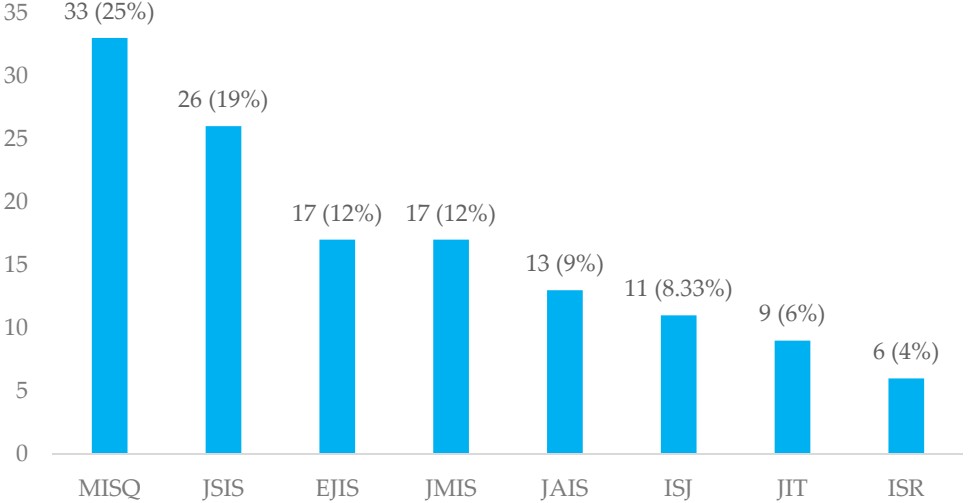

**Figure 2.** Article distribution by journal.

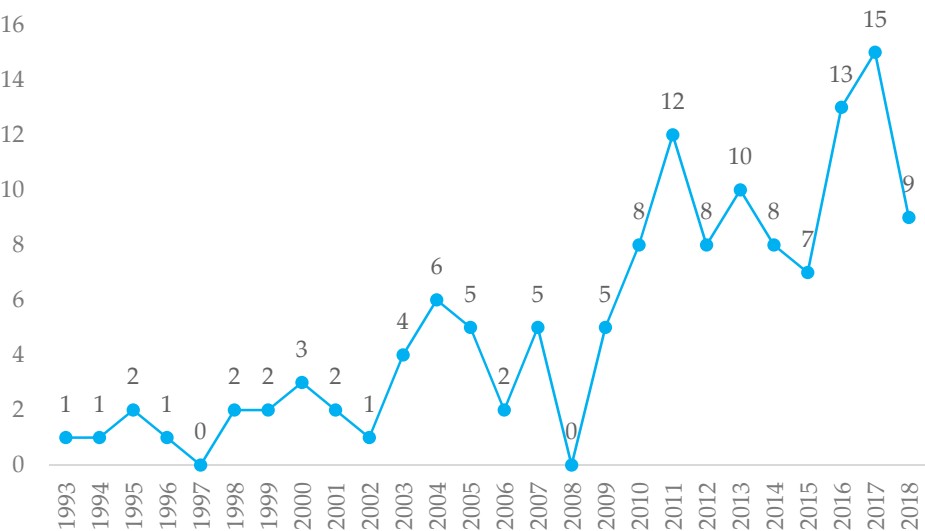

**Figure 3.** Article distribution by year of publication.

### 4.1.3. Distribution by Article Type

Different research approaches were employed in IS literature related to sustainability, but they were mainly categorized into six types of research approaches: empirical, conceptual, modelling, design science, review, experimental, meta-analysis, and simulation (Figure 4). Accordingly, empirical

studies were the most common type of article from IS literature on sustainability, while experimental studies were the least common type. After this, to understand research directions, further analysis for the 77 empirical studies was conducted with the data collection method, data analysis method, and research contexts.

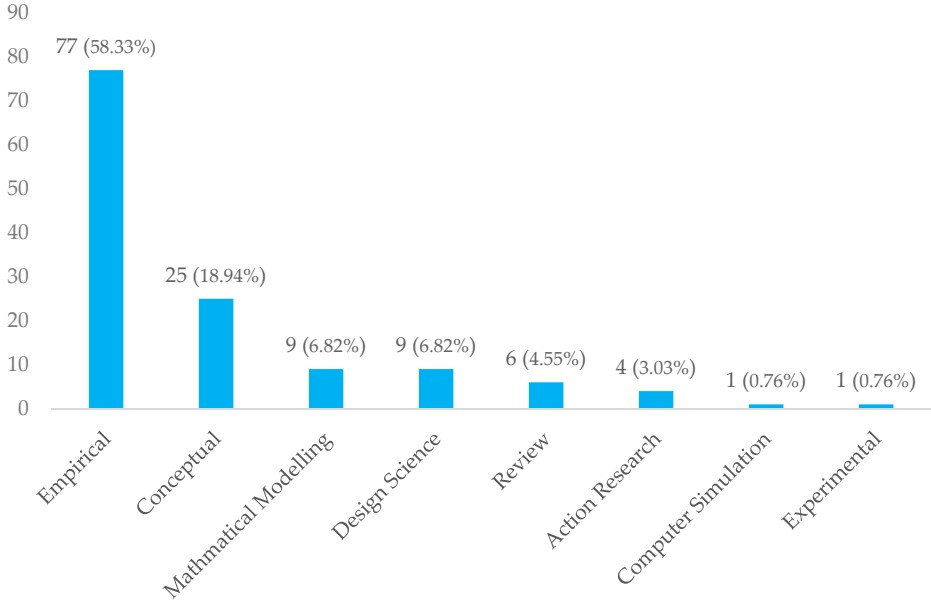

**Figure 4.** Distribution of articles by type.

### 4.1.4. Methodologies Used in Empirical Studies

Out of the 77 empirical studies, four main data collection methods were used, such as archival, multi-source, surveys, and interviews (Figure 5). Accordingly, the archival method was shown to be the predominant data collection method and interviews were the least used data collection method.

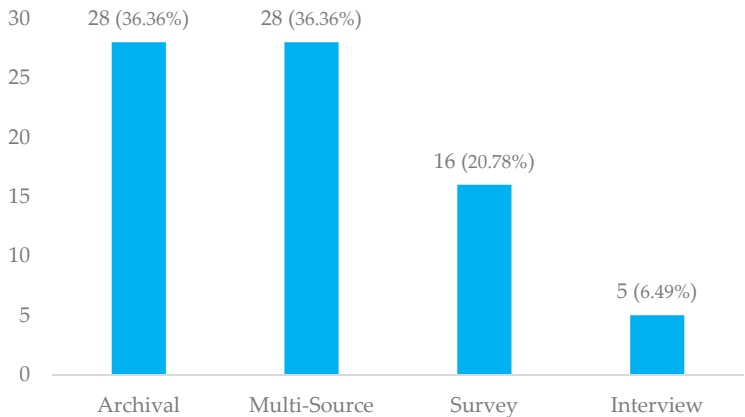

**Figure 5.** Distribution of data collection methods in empirical studies.

In terms of empirical studies, five main data analysis methods were explored, such as statistical modelling, content analysis, mixed methods, narrative case analysis, and process views (Figure 6). Accordingly, the statistical model method (i.e., regression, structural equation modelling) was shown to be the predominant data analysis method and the process view was the least used data analysis method [63].

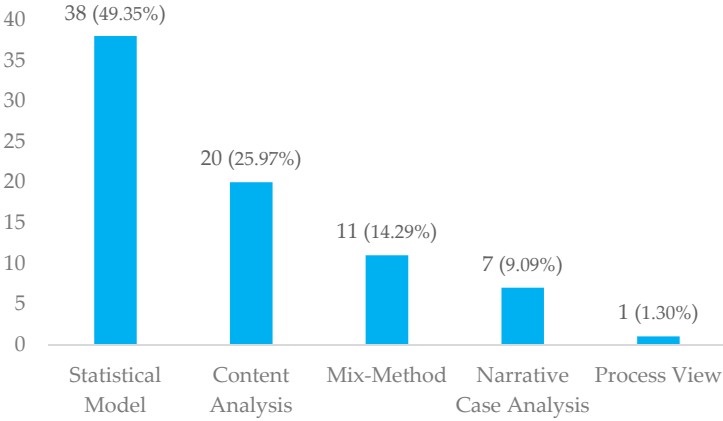

**Figure 6.** Distribution of data analysis methods in empirical studies.

### 4.1.5. Research Context in Empirical Studies

Out of the 77 empirical studies, five countries—the USA, the UK, India, Canada, and China—were predominantly examined in IS literature (Table 2). For empirical studies, 28 (36.36%) studies mostly focused on the USA, and three (3.90%) studies focused on China, respectively. This implies that a lot of the research on IS has been highly focused on developed countries. In order to study the relationship between the physical and informational components of green projects, Watson et al. [64] compared four green transportation projects from four countries (France, the US, Singapore, and Chile).

**Table 2.** Research context: Countries (N = 77).

| Country | Number (%) |
| --- | --- |
| USA | 28 (36.36%) |
| UK | 7 (9.09%) |
| India | 4 (5.19%) |
| Canada | 3 (3.90%) |
| China | 3 (3.90%) |
| Australia | 2 (2.60%) |
| Germany | 2 (2.60%) |
| The Netherlands | 2 (2.60%) |
| Malaysia | 1 (1.30%) |
| Spain | 1 (1.30%) |
| Brazil | 1 (1.30%) |
| Greece | 1 (1.30%) |
| Denmark | 1 (1.30%) |
| Singapore | 1 (1.30%) |
| Mexico | 1 (1.30%) |
| Switzerland and Germany | 1 (1.30%) |
| France, USA, Singapore and Chile | 1 (1.30%) |
| UK, Spain, Demark and the Netherlands | 1 (1.30%) |
| Not specific | 16 (20.78%) |

Besides region and country, the top five industries were studied in IT healthcare, government, manufacturing, and retail (Table 3). For empirical studies, 15 (19.48%) studies mostly focused on the IT industry, while two (3.90%) studies focused on the agriculture and transportation industries, respectively. For instance, to understand the levels of business information intensity, Griffiths and Finlay [65] compared financial services and retail manufacturing. Nishant et al. [66] explored different industries in industrial and commercial machinery, computer equipment, electronics, electrical equipment and components, business services, and communications. Interestingly, the studies that discussed the role of IS/IT in the establishment of sustainable competitive advantages were more

inclined to study companies in different industries. This is because, regardless of industry sector, findings can be similar and applicable to different industries [7,67,68].

**Table 3.** Research context: Industries (N = 77).

| Industry | Number (%) |
| --- | --- |
| IT | 15 (19.48%) |
| Healthcare | 11 (14.29%) |
| Government | 6 (7.79%) |
| Manufacturing | 5 (6.49%) |
| Retail | 3 (3.90%) |
| Agriculture | 2 (2.60%) |
| Transportation | 2 (2.60%) |
| Others | 20 (25.97%) |
| Not specific | 13 (16.88%) |

### 4.1.6. Discussion and Implications

Based on a systematic literature review, a total of 132 articles in IS literature were selected, particularly focused on sustainability. Although sustainability research has existed for quite a long time, there is a relatively limited amount of studies on the IS discipline [35,51]. Therefore, to describe the nature of sustainability in information systems-related articles since 1993, a systematic literature review was employed to analyze a total of 132 articles.

The overall finding from the systematic literature review was that most articles established research through designing empirical studies. Empirical studies were conducted to explore how IT can improve the environmental sustainability of corporate operations, and how green IS initiatives can be implemented as a part of a corporate strategy towards the organizational sustainability process [67,69]. Hu et al. [70] found that environmental awareness, industry norms, internal readiness, and customers' and equity holders' attitudes were key drivers in green IT practices. Other important influences were government regulations and their competitors. Therefore, Hedman and Henningsson [69] asserted that green IS initiatives should be a part of a corporate strategy if it is in line with the organizational agenda. Few studies identified research approaches that were explored in design science, action research, mathematical modelling, computer simulation, and experimental methods. Design science and action research, as IS-specific scientific paradigms, might be used in the future. In the past, IS scholars explored a design science approach to assess different frameworks in the area of environment sustainability. This includes the role of information systems in stimulating energy-efficient behavior in private households, the quantification of location-based investment incentives in renewable energy support mechanisms, and the sensemaking support systems in environmental sustainability transformations [71–73]. As design science provides a problem-solving paradigm that helps define ideas, practices, and technological capabilities, the use of information systems can be efficiently integrated [74–76].

On the other hand, the action research approach is a combination of practice and theory, which helps solve practical problems and expand scientific knowledge [77]. Action research could be summarized as a two-stage process including the "diagnostic" stage and the "therapeutic" stage (Blum [78], p.1). During the whole process, action researchers bring knowledge of action research and general theories, whereas clients bring practical knowledge [79]. Baskerville and Myers [79] believed that the action research approach can provide a prospective way to improve the practical relevance of IS research. Therefore, the action research approach is a powerful tool to study in the interaction between IS, humans, the environment, and society [77]. Due to recent environmental and societal problems, both the design science approach and the action research approach can help provide possible suggestions for practical business solutions, theoretical and academic, in IS [74].

Particularly with empirical studies, the archival method is predominantly used for primary data analysis in IS literature, followed by questionnaires and interviews. Numerous archival approaches

posited that managerial IT skills are positively related to sustainability, and competitors' knowledge of competitive advantage is negatively related to sustainability [7]. In addition, the articles that discussed the effect of corporate IS/IT capability on sustainable competitive advantage were more likely to collect data from widely circulated journals in the US, including those from the CompStat database and InformationWeek. This is because, since 1991, InformationWeek has identified about 40 to 50 firms (out of the 500) each year as the "leaders" in IT technology in their respective industries, and has provided data such as IT budget and the size of IT staff, which can help researchers quantify corporate IT capability [67,68].

Furthermore, most studies using the empirical approach were conducted in developed countries, such as the United States and Europe, which shows an importance to study developing countries' IS studies. This may be because issues of sustainability have been highlighted in many developed countries longer than in developing countries. This would lead to many changes and shifts in the environments and standards of operations for enterprises in developed countries. As many developing countries are becoming suppliers for developed countries or manufacturers, it is becoming increasingly important to adopt higher standards for sustainable operations. As numerous empirical studies have focused on the IT industry with less focus on manufacturing, it is important to explore how IS can improve sustainability and enhance firm performance in developing countries.

### 4.2. Results from Citation Network Analysis

Among the 132 articles, the six domains were captured as environmental sustainability, sustainable competitive advantage, sustainable online social communities, the role of IS in developing countries, sustainable technology infrastructures, and the scattered articles cluster. Out of 32 papers (24.24%), the environmental sustainability issue was shown to be the largest and most populated research domain. The second largest research domain, with 31 papers (23.48%), can be termed as "sustainable competitive advantage" followed by the third largest research domain with nine papers (6.82%) focusing on sustainable online social communities. On the other hand, two emerging research domains were captured and divided into the role of IS in developing countries, and sustainable technology infrastructures. Emerging research domains were domains which focused on similar topics but had low connectivity. One of the emerging research domains, with four papers (3.03%), was classified as "the role of IS in developing countries." Another emerging research domain, with three papers (2.27%), was referred to as "sustainable technology infrastructures." On the other hand, the 53 papers that could not be included in any research domain (40.16%) were classified as the "scattered articles" cluster. Detailed information is illustrated in Figure 7.

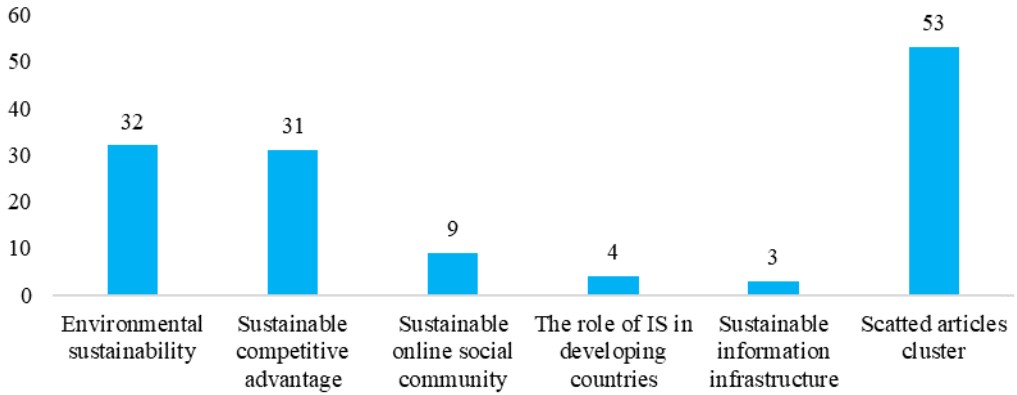

**Figure 7.** Research domain classification.

Interestingly, it was found that the sustainable competitive advantage research domain had associations with the environmental sustainability research domain and sustainable online social communities, respectively (Figure 8). Benitez-Amado and Walczuch [80] stated that information

technology capabilities are able to promote environmental strategies, which would be a significant moderator of the effects of information technology on firm performance. Meanwhile, Dao et al. [81] indicated that focusing on developing sustainability capabilities might not only serve the environment and people, but also help enterprises generate value that could enhance profitability and provide a sustained competitive advantage. Among the connections between the sustainable competitive advantage research domain and sustainable online social communities, Wade and Hulland [33] was related to Butler [82], as both studies explored from the perspective of the resource-based view.

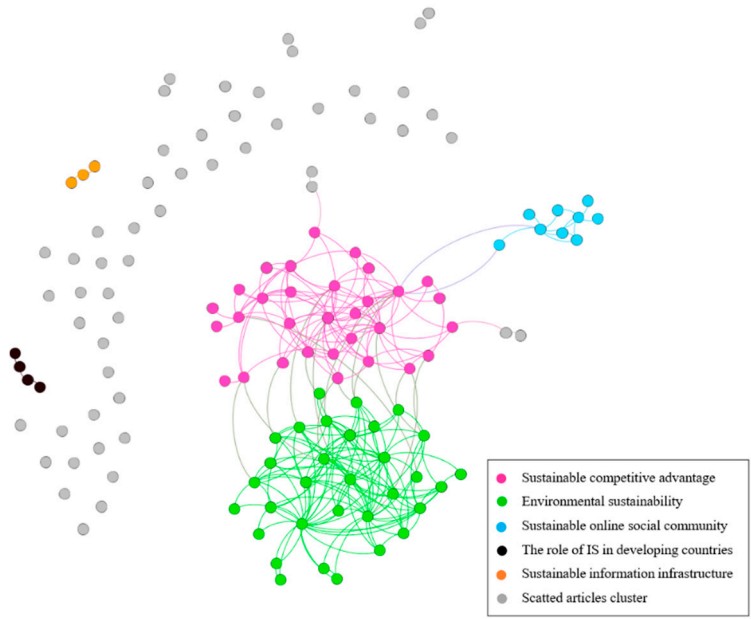

**Figure 8.** Research domain classification of sample publications.

*4.3. Results from MPA*

4.3.1. Results from Main Clusters

To further examine the knowledge structure of the three major research domains (i.e., sustainable competitive advantage, green IS, and sustainable online social communities), a main path analysis was conducted through weighting the citations in the cluster to identify the most important citation path [39].

Sustainable Competitive Advantage

The sustainable competitive advantage was captured through the main path analysis (Figure 9). The majority of articles used the resource-based theory (RBT) as the research framework to investigate the relationship between information systems and sustainable competitive advantage [33,34,67,83–86]. The RBT defines firm resources as "all assets, capabilities, organizational processes, firm attributes, information, knowledge, etc. controlled by a firm" ([87], p. 101). Two conditions have to be realized to achieve a sustainable competitive advantage: 1) maintaining resources that are valuable and rare, and 2) preventing competitors from replicating processes [33,85]. Therefore, RBT has predominantly been used in IS research because IS are positioned as one of the strongest resources and as a corporate strategy which can improve firm performance [33]. The key contribution of each paper is shown in Table 4.

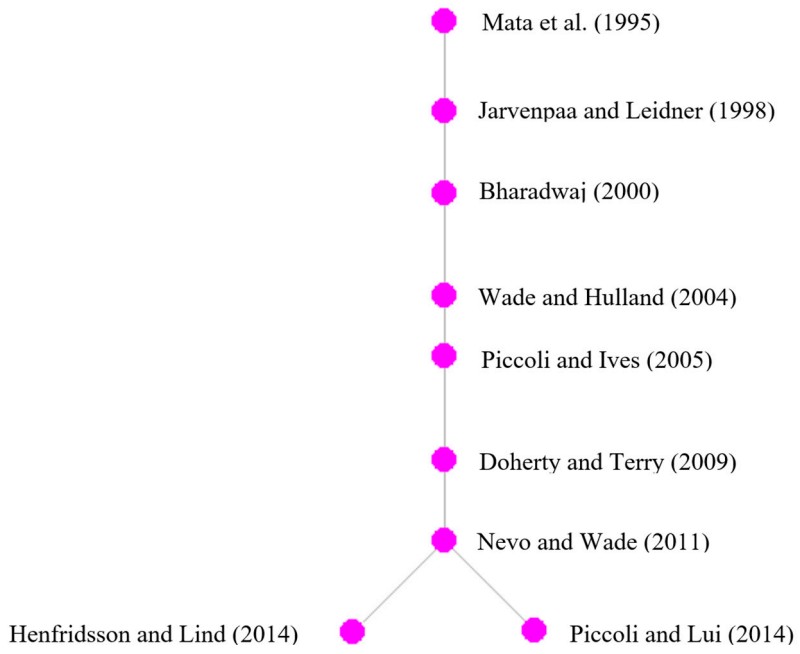

**Figure 9.** Main path of sustainable competitive advantage.

**Table 4.** The key contribution of each paper.

| Authors | The Kinds of IS Resources Affecting Sustainable Competitive Advantage |
|---|---|
| Mata et al. [83] | Managerial IT skills |
| Jarvenpaa and Leidner [84] | · The role of social and governmental relationships outside the enterprise<br>· Dynamic capabilities (i.e., strategic vision, flexibility) |
| Bharadwaj [67] | Higher IT capability with a combination of IT infrastructure, human resources, and IT-enabled intangibles |
| Wade and Hulland [33] | · "Inside-out" resources<br>· "Spanning" resources<br>· "Outside-in" resources |
| Piccoli and Ives [34] | · IT resources barrier<br>· Complementary resources barrier<br>· IT project barrier<br>· Pre-emption barrier |
| Doherty and Terry [85] | · "Outside-in" IS resources<br>· "Spanning" IS resources |
| Nevo and Wade [86], Henfridsson and Lind [88], Piccoli and Lui [89] | Synergies between organizational resources and IT assets |
| Future direction | Integrate the concept of complexity and dependency to explain the relationship between IT assets and complementary resources |

In the 1990s, studies in IS for sustainable competitive advantage were focused on a firm management's ability to improve and exploit IT applications to support and enhance other business functions, because competitors can easily buy or replicate tangible IT resources. In addition to focusing on managing IT within an enterprise, Jarvenpaa and Leidner [84] have suggested the role of social and governmental relationships outside the enterprise to enable enterprises to improve sustainable competitive advantage via IS. Jarvenpaa and Leidner [84] found that dynamic capabilities, including

strategic vision and flexibility, coupled with trusted core capabilities, could be key factors influencing internal and external changes in an unstable environment which lacks an IT infrastructure and culture [84]. Similarly, Bharadwaj [67] pointed out that enterprises could achieve better performance when they have higher IT capability with a combination of IT infrastructure, human resources, and IT-enabled intangibles. For instance, a strong top management commitment to IS could possibly help develop IS resources through isolating mechanisms such as time compression diseconomies, resource connectivity, and social complexity [33,67]. In the context of the strategic initiatives that rely on IT, Piccoli and Ives [34] name the four erosion barriers as the IT resources barrier, complementary resources barrier, IT project barrier, and pre-emption barrier, that support IT through response-lag driving factors independently or in combination with each other to maintain competitive advantage [34]. The response-lag drivers of IT resource barriers encompass IT capabilities (i.e., technical skills, management skills, relationship assets) that link to the complementary strengthening of resource barriers through IT-dependent strategic initiatives (i.e., top management commitment, corporate culture) [34].

Wade and Hulland [33] focused on the use of three enterprises' resources, such as inside-out, spanning, and outside-in, to enhance competitive advantage. Inside-out IS resources refer to how responsive firms respond to the market requirements and opportunities via technological development and cost controls, while outside-in IS resources strive to achieve market responsiveness, managing external relationships to enhance consumer loyalty, and to understand competitors [33]. Spanning IS resources combine inside-out IS resources and the outside-in IS resources that are considered in the management of IS/business partnerships and IS management and planning [33]. Based on the findings of Wade and Hulland [33] study, outside-in and spanning IS resources were shown to have more influence than inside-out IS resources on either initial competitive advantage or long-term competitive advantage [33]. When firms reside in stable business environments or high munificent and complex environments, inside-out resources may have a higher impact on firm performance. Otherwise, outside-in resources and spanning resources have a stronger impact on firm performance with firms in turbulent business environments and high complexity environments [33]. A later study by Doherty and Terry [85] further extended Wade and Hulland's [33] study to posit that implementing IS initiatives to successfully achieve sustainable competitive advantages may depend on the "outside-in" and "spanning" IS resources.

To explore sustainable competitive advantage, Nevo and Wade [86] extended the studies of Bharadwaj [67] and Wade and Hulland [33]. If IT assets and organizational resources are properly combined in a particular context as value, rarity, appropriateness, imitability, sustainability, and imperfect mobility, it is possible to create positive synergetic results [86]. A recent study by Nevo and Wade [86] found that the value, rarity, and inimitability of the resultant IT-enabled resource created better synergetic impact, increasing positive firm performance. That is to say, value and rarity may influence operational benefits, while value and inimitability drive stronger strategic benefits [86]. Similarly, Piccoli and Lui [89] supported Nevo and Wade's [86] view by suggesting that synergies between organizational resources and IT assets may play an important role in developing the competitive impact of IT-dependent strategic initiatives. This view was carried out in Henfridsson and Lind's [88] study by using IS to implement sustainability strategies that can become a competitive advantage for a variety of organizational sub-communities. Therefore, the interaction between IT assets and complementary resources may be established as features that are only held via IT-dependent strategic initiatives. This could lead future research used to integrate the concept of complexity and dependency to explain the relationship between IT assets and complementary resources [89].

Environmental Sustainability

IS can be important resources in achieving environmental sustainability, but environmental sustainability was not discussed in IS literature until the late 2000s [35] (Figure 10). This may be because people lack a universal understanding of the potential impact of climate change and the salient factors needed to achieve environmental sustainability [19]. Watson, Boudreau, and Chen [90]

addressed how energy informatics can contribute to environmental sustainability. Considering IS as an organizational change process which assists belief formation, sustainability actions, and environmental and organizational performance outcomes, Melville [35] established a belief–action–outcome (BAO) framework linking macrostructure (society, natural environment, organization) with micro-structure (individual) to describe three aspects of sustainable beliefs that are regulated by IS. Thus, organizational and individual actions to enhance sustainability practices can be motivated by sustainable beliefs, and both environmental and financial performance outcomes can be brought through actions. The key contribution of each paper is shown on Table 5.

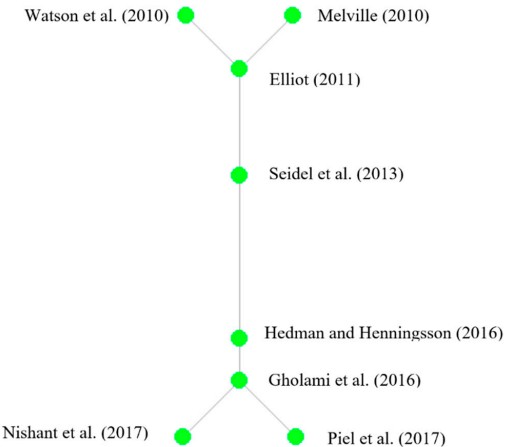

**Figure 10.** The main path of environmental sustainability.

**Table 5.** The key contribution of each paper.

| Authors | Key Contributions |
|---|---|
| Watson et al. [90] | How IS academia contributes to environmental sustainability from four aspects: research, education (i.e., develop energy informatics), journals and associations |
| Melville [35] | The factors that promote or inhibit the adoption of environmentally sustainable business practices (i.e., culture), the relationship between environmentally sustainable business practices and business performance, and the relationship between IS and environmental performance of supply chain |
| Elliot [19] | · The challenges to environmental sustainability including accessing the state of environmental deterioration, the acceleration to deterioration caused by human activities, and the uncertainty of human response to deterioration.<br>· How people face the challenges through reviewing previous studies in the areas of environment, society, governments, industries and alliances, organizations, and individuals and groups within organizations. |
| Seidel et al. [91] | Key affordances of green IS enabling organizational transformation through organizational sense-making and sustainable practices.<br>· Reflective disclosure<br>· Information democratization<br>· Output management<br>· Delocalization |
| Hedman and Henningsson [69] | Green IS initiatives could contribute as one type of organizational projects if it aligned with the organizational agenda |
| Gholami, Watson, Hasan, Molla and Bjorn-Andersen [74] | Future IS research on environmental sustainability has to position itself as more solutions-oriented and has to utilize IS knowledge in more practical and innovative ways to tackle environmental and social issues. |
| Nishant, Teo and Goh [66] | Green IT had a positive impact on market returns which could provide better assurance for the long-term performance of investments |
| Piel, Hamann, Koukal and Breitner [72] | IS can provide solutions to inform policy decisions, and integrate intermittent renewable energy sources |
| Future direction | · Provide more solutions for environmental sustainability issue<br>· Use sentiment analysis to measure the response of all stakeholders to green IS initiatives<br>· Why do some green IS initiatives have high abnormal returns in stock market |

Numerous studies [19,69,72] studied how IS enhanced the organizational transformation towards environmental sustainability. Elliot [19] emphasized how human behavior has changed toward the environment by discussing how the relationship between human beings and technology can improve environmental sustainability. Seidel, Recker, and vom Brocke [91] suggested providing a way to understand changes in sustainability themes (organizational sense-making) and to initiate work practices that are in line with sustainability goals (sustainable practicing). Therefore, they explained the key benefits of green IS as the enabling of organizational transformation through organizational sense-making and sustainable practices. The main benefits of green IS encompass reflective disclosure (allowing monitoring, analysis of the performance of current practices), information democratization (all the individuals in the organization can access and use the information related to sustainability from multiple sources), output management (directing individuals' work actions to follow certain boundary conditions to reduce negative environmental impact), and delocalization (changing work practices to become less location-dependent and consequently reducing the negative impact from resource movement) [91].

Hedman and Henningsson [69] established a green IS organizational response model to show the interaction between green IS initiatives and the organizational sustainability process. Green IS initiatives could contribute as one of the types of organizational projects if it aligned with the organizational agenda [69,92]. In this way, green IS projects can influence future green IS initiatives and implement a better sustainability process. However, Gholami, Watson, Hasan, Molla, and Bjorn-Andersen [74] argued that future IS research on environmental sustainability has to position itself as more solutions-oriented and has to utilize IS knowledge in more practical and innovative ways to tackle environmental and social issues. This view was supported by Gholami, Watson, Hasan, Molla, and Bjorn-Andersen [74] and Piel, Hamann, Koukal, and Breitner [72] which suggests that IS can provide solutions to inform policy decisions, and integrate intermittent renewable energy sources. Likewise, a lot of literature suggests that design science research can guide the future research direction of environmental sustainability in IS [73,93]. A recent study by Nishant, Teo, and Goh [66] found that green IT had a positive impact on market returns which could provide better assurance for the long-term performance of investments. Similar to Nishant, Teo, and Goh's [66] findings, Loeser et al. [94] also suggested that it is important to explore how environmental sustainability can impact economic sustainability. To establish community-driven environmental sustainability, Tim et al. [95] strongly suggested that social and environmental sustainability go hand in hand.

Based on IS literature on environmental sustainability, previous studies focused on how IS have contributed to solving environmental issues and problems in human society. However, more studies on environmental sustainability in IS literature were found to focus on enterprises' environmentally sustainable transformations in later literature. It is important to pinpoint how environmental sustainability is not only looking into business practices to use for short-term performance, but rather to position itself as a crucial practice in IS literature.

Sustainable Online Social Communities

As one of the emerging areas, articles on sustainable online social communities constitute the third largest research domain [96]. Online communities are highly accessible and are self-rising, decentralized IS [97]. IT offers a technological infrastructure for potential social activities, but it does not guarantee that individuals will participate in such online social activities [82]. If a social structure cannot exceed individuals' expectations, it may be unable to attract new members and thus become unsustainable [82,98]. Therefore, promoting participation in an online community is an important issue in IS literature [82,96,98–101]. The development path of sustainable competitive advantage cluster is shown on Figure 11 while the key factors affecting sustainable online social community in concluded and shown on Table 6.

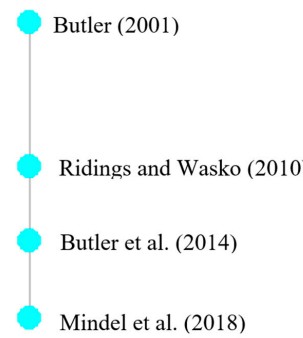

**Figure 11.** The main path of sustainable online social communities.

Based on members' time, energy, and other resources, online social communities attract and retain members that can provide benefits including information and social support [82]. Accordingly, Butler [82] found that membership size and communication activities in online social communities had an influence on sustainability activities. More specifically, the influence of membership size on an online social structure to attract and retain members can be mediated by communication activity. To some extent, constructing a community is a process that collects resources provided by members, and communities need to attract a large pool of members, as this is the main source of benefit [82]. Therefore, membership size is used to measure resource availability [82]. Likewise, Ridings and Wasko [98] argued that it is necessary to investigate the effect of the reciprocal interactions between the structural dynamics (i.e., membership size, communication activity) and social dynamics (i.e., the interactions among members). On the other hand, Jones et al. [102] suggested that when information overloading hinders advantages and benefits, one coping strategy to deal with continuously changing communication behavior is to leave the community [102]. In other words, more resources do not always lead to better communication activities, but they can allow for social interaction between members to preserve the community. Moreover, as individual efforts increase, the retained members also increase, but communities may sacrifice their ability to attract new members in the process [96].

**Table 6.** The key factors affecting sustainable online social community.

| Authors | Key Factors |
|---|---|
| Butler [82] | · Membership size<br>· Communication activity |
| Ridings and Wasko [98] | · Basic nature of interaction among members<br>· Relationship between members |
| Butler et al. [96] | · participation costs<br>· topic consistency cues |
| Mindel et al. [97] | · Provision<br>· Appropriation<br>· Revitalization<br>· Equitability |
| Future direction | The difference between types of stakeholders' (i.e., producer, provider and appropriator) actions in an online community |

Butler [82] only focused on the effect of structural dynamics (i.e., membership size) on community sustainability, while Ridings and Wasko [98] combined structural dynamics and social dynamics (i.e., the interactions among members) when examining community sustainability. By extending the findings of Butler [82] and Ridings and Wasko [98], Butler, Bateman, Gray, and Diamant [97] highlighted how technology that changed users' participation experiences affected the sustainability of online social communities due to the effects on members' behavior and, thus, the function of the

community. By examining the influence of the cost of participation (e.g., how much time and effort is required to engage with the content provided in a community), and topic consistency cues (e.g., how strongly a community signals that the topics that may appear in the future will be consistent with those it has hosted in the past), Butler, Bateman, Gray, and Diamant [96] found that there is a curvilinear relationship between community resilience and topic consistency cues. More specifically, community resilience was greater when community platforms presented low and high topic consistency cues, while it became low when platforms showed moderate topic consistency cues [96].

Mindel, Mathiassen, and Rai [97] integrated the empirical evidence and theoretical contributions from previous literature on the sustainability of online communities. They reviewed 73 studies and developed a global view on how online communities mitigate the impact of their high degree of openness to sustainable participation. They proposed a new mode of governance, "polycentricity", because people usually communicate together to find a sensible way of sharing resource, rather than depending on central governance. Generally, in online social communities, stakeholders are producers (the architects and sponsors of the infrastructure that enable the system), providers (the people who provide information to the system), and appropriators (the people who extract information from the system) [103]. A sustainable online community allows stakeholders to continuously derive value [97]. Feedback on value derived by stakeholders affects the evolution of an online community's polycentric governance practices [97]. In addition, sustainability is constructed by provision, appropriation, revitalization, and equitability [97]. Sustainable online communities are premised on a system of complements constructed by continuous provision and appropriation as well as revitalization and equitability [97]. Online social communities are highly accessible to both content consumers and producers, so individuals can join for free and leave at any time, which may lead to high fluctuations in consumer and producer participation [97]. Therefore, online communities usually operate under conditions of uncertainty that make them more vulnerable to collective action threats, including free-riding, congestion, pollution, violation, and rebellion [97]. Individual providers and appropriators' collective action threats create vulnerabilities to the sustainability of an online community [97]. Polycentric governance practices of boundary regulation, shared accountability, incremental adaptation, and provider recognition can reduce collective action threats and increase the sustainability of an online community [97]. In the future, Mindel, Mathiassen, and Rai's [97] findings can be applied to study organizational IS, such as open-access enterprise-sponsored systems and crowdsourcing initiatives created for soliciting ideas or specific tasks from information providers.

### 4.3.2. Unclassified Publications

#### Emerging Cluster

Two emerging research domains were captured through the main path analysis. The first emerging research domain focused on how IS help developing countries, including establishing health information systems [104,105], and telecentres [106]. The second emerging research domain focused on the technology infrastructure for sustainable IS [107–109].

#### The role of IS in Developing Countries

The first emerging domain by Srivastava and Shainesh [63] related to how IS can assist the convergence of developing countries by connecting the knowledge streams from two different sources, such as from Madon [106], and Braa, Monteiro, and Sahay [104] (Figure 12). One of the sources from Madon (2005) and Srivastava and Shainesh [63] emphasized the digital divide as an unbalanced distribution of Information Communication Technology (ICT) resources amongst different societal groups. Using the example of the Akshaya project in Kerala, Madon [106] identified five key issues in the sustainability of telecentres, which included building corporate confidence, working with the government, renewing grassroots campaigning involving the legislative system, and continuing the support of political champions. However, Madon's [106] dominant viewpoint considered telecentres

valuable only in which (1) governments have sufficient resources to provide the necessary digital goods, and (2) society is capable enough to transform the digital goods into desired outcomes. Therefore, Srivastava and Shainesh [63] argued that this view may have constrained applicability in developing countries, where the majority of the population does not have accessibility to basic capabilities, such as healthcare and education. This suggestion was anchored in a service-dominant logic, which investigates how to bridge the digital divide in developing countries. From a service-dominant perspective, ICTs can help bridge the service divide to enhance the capabilities of service-disadvantaged segments of society. However, such service delivery requires the innovative assembly of three interactional resources, such as knowledge, technology, and institutions [63]. In addition, Srivastava and Shainesh [63] extended the research agenda from Braa, Monteiro, and Sahay [104] and Braa, Hanseth, Heywood, Mohammed, and Shaw [105] in respect to the usage of ICT in the healthcare sector in developing countries. More particularly, Braa, Monteiro, and Sahay [104] explored the sustainable health information system program across the developing countries by looking at political support, the development of health information systems, and training and education. Braa, Hanseth, Heywood, Mohammed, and Shaw [105] further proposed the concept of flexible standards as a key element in a sustainable infrastructure development strategy and highlighted the importance of developing flexible standards for health information systems in developing countries.

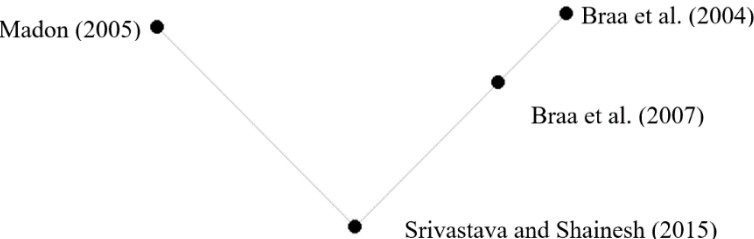

**Figure 12.** The research domain of the role of is in developing countries.

The role of IS in Developing Countries

The second emerging domain relates to sustainable information infrastructure divergence, in which Ribes and Finholt's [107] study lays the foundation for the studies of Ribes and Polk [108] and Venters, Oborn, and Barrett [109] (Figure 13). Sustainable infrastructure is intended as a long-term invisible support structure [107]. Maintaining the sustainability of an information infrastructure can be challenging in its planning, design, implementation, and maintenance [107]. Previous studies in the field of information infrastructure primarily focused on socio-technical concerns, which neglect the importance of circumstances and the particularities of change in which infrastructures must be flexible [107]. Thus, Ribes and Polk [108] proposed three areas—socio-technical, technoscientific, and institutional—to match the forms of flexibility to the heterogeneity of changes an infrastructure may encounter. Ribes and Polk [108] suggested technoscientific flexibility as a form of adaptability and resilience that is, perhaps, most central to a research infrastructure, in that it supports scientific research and the dramatic transformations objects that research undergoes [108].

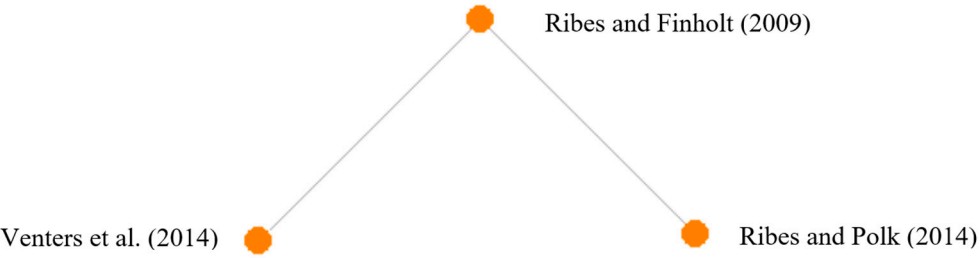

**Figure 13.** The research domain of sustainable information infrastructure.

On the other hand, Venters, Oborn, and Barrett [109] addressed how emerging tensions can extend Ribes and Finholt's [107] study on information infrastructure development and evolution. At different temporal dimensions, Venters, Oborn, and Barrett [109] identified three coordination tensions, such as obtaining adequate transparency in the present, modelling an infrastructure in future, and the disciplining of social and material inertias in the past. In addition, Venters, Oborn, and Barrett [109] emphasized the dynamic aspects of sustainable information infrastructure, which implies social and material agencies that enact information infrastructure in multiple dimensions of time in a dynamic interplay. Therefore, the sociometrical approach of Venters, Oborn, and Barrett [109] established an innovative perspective for future studies on digital infrastructures and their coordination.

Scatters

Lastly, another research domain with unclassified articles was captured, in which the articles focused on similar topics, but featured references that were not connected to the other articles. A majority of these articles discussed how enterprises enhanced sustainable competitive advantage through IS/IT-enabled tactics, such as knowledge management [110], customer analysis based on big data [111], and social commerce [13]. Given that IS play a significant role in inter-organizational collaboration, Kumar and van Dissel [112] analyzed the basic features (i.e., configuration, coordination mechanism, etc.) of three types of interorganizational systems, such as information resource interorganizational systems, value/supply chain interorganizational systems, and networked interorganizational systems. More specifically, studies for particular inter-organizational systems examined e-marketplaces [113], supply chain collaboration systems [12], and inter-organizational knowledge sharing [114]. Few studies examined the sustainable business model for IT companies such as online dataset services [115], game companies [116], mobile apps [117], mobile data services [118], and telemedicine services [119].

## 5. Discussion and Limitations

Due to increasing social issues regarding sustainability in many different areas, this study conducted a systematic citation network analysis on how the sustainability issue is discussed in IS literature. This study is one of the few that reviews studies to adopt the SLNA research method in the IS discipline. As SLNA integrates the advantages of SLR and CNA, SLR was used to investigate the nature of sustainability in IS, whereas CNA helped to locate the main research issues and analyze the dynamic knowledge evolution within each research issue. A total of 132 articles were captured from eight leading IS journals that focused on sustainability from the period 1993 to 2018. Out of the eight leading IS journals, most articles were published in *MISQ*, and empirical study was recognized as the primary research method. Although a majority of previous studies focused on exploring developed countries, there has been a dramatic shift towards examining developing countries, as those countries are becoming more important players in global business operations such as production and manufacturing.

Besides the trends in IS literature in the area of sustainability, it was found that three research domains and two emerging research domains were highly dominant among the 132 articles. As IS are used to manage businesses in their initial stages, the earlier literature tended to discuss how IS can help enterprises achieve a sustainable competitive advantage through developing their competitive resource (capability) and the barriers to erosion from the perspective of the RBT. More recently, it was suggested that competitive resources could create a synergy between information systems and other corporate resources, rather than IS as tangible assets or managerial skills. In the past, people believed that IT technology or managerial IT skills were fundamental to maintaining the sustainable competitive advantage of enterprises. However, it was found that the synergy between IS and the other resources within the company had a positive effect on its sustainable competitive advantage. It is worth noting that the synergy between IS and other resources can ultimately meet customers' needs, whether through more effective operations, mass customization, or new products [120]. For instance, Aeroflot, a Russian airline company, constructed information systems to significantly improve their operations.

Their information systems not only provided management with an instant overview of more than 450 key performance indicators, but also information about aircraft performance, and even preventive maintenance [120]. Therefore, to successfully implement IS, companies must also pay attention to customer demands, operational flexibility, and the cultivation of an innovative culture [120].

In the domain of sustainable online social communities, important ways that can influence the sustainable development of online social communities were found to be structural dynamics (i.e., membership size, communication activity) [82], social dynamics (the basic nature of interaction among members and the relationship between members) [96], and technology factors (i.e., participation cost and topic consistency) [97]. For organizational online communities, polycentric governance practices can be used to mitigate collective action threats and keep the sustainability of online communities [97]. In the era of Web 2.0, enterprises increasingly participate in interactions with consumers through online communities [121]. Enterprises should thus consider how to communicate with their consumers in online communities to promote sustainable competitive advantage. Lee, Lee, and Oh [13] investigated how social reference systems, such as Facebook "likes," were used to promote sustainable financial performance in social commerce. Enterprises can also disseminate their ideas for environmental sustainability [121]. Overall, IS literature is highly focused on three areas: economy, environment, and society, which supports Elkington's [36] Triple Bottom Line theory.

Although the findings of this study are valuable to IS literature, there are some limitations that can be improved in future studies. First, to focus on IS literature, the leading eight IS journals were selected. However, this limits how sustainability in IS literature is being presented in academia from different disciplines. Future research can be extended to a wider pool of journals to truly understand how the IS literature is changing the trends of knowledge and to learn about the future direction of enterprises utilizing IS in their systems. Second, this study collected 132 articles to understand the trends in IS literature. Although this is an adequate sample size for main path analysis and CNA, expanding the sample size would help in comprehending the trends in IS literature. Third, the findings of this study, using a keyword search, were insightful for determining the direction of academia and enterprises in IS. Therefore, it would be better to explore IS studies with more keywords related to sustainability, supply chains, and agile logistics in IS literature.

**Author Contributions:** F.Z. conceived this research and wrote the manuscript. S.H.N.L. and C.K.Y.L. commented for the entire study and provide insightful suggestions. All authors have read and agreed to the published version of the manuscript.

**Funding:** This research was partially supported by the Research Grants Council of Hong Kong under grant number PolyU ZJM0.

**Conflicts of Interest:** The authors declare no conflict of interest.

## Appendix A

**Table A1.** Classification results

| Research Domain | Articles |
|---|---|
| Sustainable competitive advantage | Dehning and Stratopoulos [7], Kettinger, Grover, Guha and Segars [10], Peppard and Ward [31], Wade and Hulland [33], Piccoli and Ives [34], Griffiths and Finlay [65], Bharadwaj [67], Mithas and Rust [68], Benitez-Amado and Walczuch [80], Mata, Fuerst and Barney [83], Jarvenpaa and Leidner [84], Doherty and Terry [85], Nevo and Wade [86], Henfridsson and Lind [88], Piccoli and Lui [89], Hedman and Kalling [92], Hidding [122], Peppard et al. [123], Ravichandran and Lertwongsatien [124], Gable [125], Gordon et al. [126], He [127], Iyer et al. [128], Lim et al. [129], Nevo and Wade [130], Atkins [131], Madon [132], Grover et al. [133], Nan and Tanriverdi [134], Rai et al. [135], Feller et al. [136] |
| Environmental sustainability | Elliot [19], Melville [35], Bengtsson and Agerfalk [51], Watson, Boudreau, Chen and Sepulveda [64], Nishant, Teo and Goh [66], Hedman and Henningsson [69], Hu, Hu, Wei and Hsu [70], Loock, Staake and Thiesse [71], Piel, Hamann, Koukal and Breitner [72], Seidel, Kruse, Szekely, Gau and Stieger [73], Gholami, Watson, Hasan, Molla and Bjorn-Andersen [74], Dao, Langella and Carbo [81], Watson, Boudreau and Chen [90], Seidel, Recker and vom Brocke [91], Brandt, Feuerriegel and Neumann [93], Loeser, Recker, vom Brocke, Molla and Zarnekow [94], Tim, Pan, Bahri and Fauzi [95], Petrini and Pozzebon [137], Butler [138], Cooper and Molla [139], Corbett [140], Hanelt et al. [141], Hasan et al. [142], Ketter et al. [143], Corbett and Mellouli [144], Abbas et al. [145], Pitt et al. [146], Bose and Luo [147], Chan and Ma [148], DesAutels and Berthon [149], He et al. [150], Marett et al. [151] |

**Table A1.** *Cont.*

| Research Domain | Articles |
|---|---|
| Sustainable online social community | Butler [82], Butler, Bateman, Gray and Diamant [96], Mindel, Mathiassen and Rai [97], Ridings and Wasko [98], Bock, Ahuja, Suh and Yap [99], Guo, Guo, Fang and Vogel [100], Phang, Kankanhalli and Tan [101], Chengalur-Smith et al. [152], Chen et al. [153] |
| The role of IS in developing countries | Srivastava and Shainesh [63], Braa, Monteiro and Sahay [104], Braa, Hanseth, Heywood, Mohammed and Shaw [105], Madon [106] |
| Sustainable information infrastructure | Ribes and Finholt [107], Ribes and Polk [108], Venters, Oborn and Barrett [109] |
| Scatted articles cluster | Hadaya and Cassivi [12], Lee, Lee and Oh [13], Kloor, Monhof, Beverungen and Braaer [30], Zyngier and Burstein [110], Kitchens, Dobolyi, Li and Abbasi [111], O'Reilly and Finnegan [113], Loebbecke, van Fenema and Powell [114], West [115], Roquilly [116], Lee and Raghu [117], Kankanhalli, Ye and Teo [118], Peters, Blohm and Leimeister [119], Assimakopoulos and Tsiligirides [154], Adomavicius et al. [155], Anderson et al. [156], Bapna et al. [157], Briggs et al. [158], Brown et al. [159], Cho et al. [160], Demirezen et al. [161], Epstein [162], Gaskin et al. [163], Gleasure and Feller [164], Grimsley and Meehan [165], Gupta and Zhdanov [166], Huang et al. [167], Iacovou [168], Jha et al. [169], Kartseva et al. [170], Levi et al. [171], Li et al. [172], Ma and McGroarty [173], Parker and Weber [174], Rajao and Marcolino [175], Ramarapu and Lado [176], Rishika et al. [177], Ruth [178], Saeed et al. [179], Sandeep and Ravishankar [180], Santos et al. [181], Saravanamuthu [182], Shaw [183], Thies et al. [184], Tyworth [185], Vaast et al. [186], Van Slyke et al. [187] |

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
