# Peer review of "The Role of Information Systems in the Sustainable Development of Enterprises: A Systematic Literature Network Analysis"

_sustainability, doi:10.3390/su12083337_

Round 1

Reviewer 1 Report

It is very good to read the paper. I appreciate the statistical work which was done.

It gave a very good overview on existing studies in this field and also I appreciate the approach of connection the Information Systems and sustainability (in link within supply chains).

Regarding the Table 1 and the suggestion of AIS and the following text: After your study, do you have additional recommendations for other journals or conferences, etc.?, just questioning. Maybe some more suggestions were found during your research.

Author Response

Dear reviewer,

Thank you for your question. Maybe we can read some Operations Management journals such as Production and Operations Management.

Best wishes,

Fan Zeng

Reviewer 2 Report

Dear author(s),

I am very glad to be given the opportunity to read this paper. The paper presents a systematic literature review analysis for sustainability in IS literature. There are three main objectives given. Appropriate tools are used, the discussion and conclusions are exhaustive.

Two minor issues:

Abstract: Abbreviations and citations should be omitted from the abstract.

Line 74: The same value annotation should be used. DT not explained.

Author Response

Dear reviewer,

Thank you for your kind comments. We have revised our manuscript according to your comments. Please kindly find the details below. Thank you.

Abstract: Abbreviations and citations should be omitted from the abstract.

→Thank you for your comment. Abbreviations and citations are removed in abstract. Please see line 11 and 28.

Line 74: The same value annotation should be used. DT not explained.

→Thank you for your comment. DT is changed to IT. Please see line 64.

Best wishes,

Fan Zeng

Reviewer 3 Report

Dear Authors

Your article is very interesting, and I am grateful for the opportunity to read it. I think that the idea and subject of the analysis is very interesting, and the results of the research give a lot of new information and possibilities of further analysis.

Reviews:

The article deals with an interesting issue and provides a new perspective on the role of IT systems in the sustainable development of enterprises.

After reading your article, I found it very interesting and I think it is an article that should be published in the journal ‘Sustanbility’. Reading the text, I found several elements that I think would improve your article.

I have 2 main comments:

No. 1. An important aspect that requires your attention is the fact that you look for similarities or differences to a lesser extent and more on presenting the content of the analyzed articles. I believe that this article should above all present the effect of your analysis. The key should be to look for common elements and differences in approaches presented by the authors of the articles you analyze. In your case, apart from the division mentioned at the beginning - this division results from the literature and not from the results of your analysis. I have not found clear elements presenting your observations or assessments. In the case of such analysis, it is important that you present the effects of your work that you have done.

No. 2. The second remark concerns the order of explanation. The scientific article has a structure that allows the reader to better understand the topic and results of your research. Your article changes the known and understood structure and logic of the statement. In my opinion, the structure of the article should be changed to meet the requirements of journal.

And detailed comments:

  1. Abstract does not discuss important elements such as the aims (Why do you made this review?), objectives, method, results - it is important not only for the reader, but also for the authors, if someone is looking for information, to know what is paper about he reads the abstract.
    Additionally:

- for example, you use the phrase "Triple Bottom Line theory" only in abstract and ending.

- The last sentence of the abstract is not related to the content of the abstract and the content of the article.

  1. The problem that I have is to understand the aim, the reason of your analisys. I think this lack of main idea/ main goal made this article difficult to understand. So, my proposition is to show it explicite in Introduction. You should start from the explication of your idea / your vision. You know it, you understand it, but I don’t. You don't just do a review just to do a review.
  2. The structure of the article have to be adjusted to the standard design. The following structure is desired:
  3. Introduction - The introduction should provide not only a research background but also an indication of a precise research/review objective, the research questions and methods used. Adding a brief indication of the logic of presenting the research material (i.e. the brief description of the content of each section of the paper) in the last paragraph of the introduction is highly recommended.
  4. Literature review - The literature review presents important highlights on the current state of the art. The content should be the starting point for further analysis. Your analysis is to develop what is at this point.
  5. Methods - the section “Methodology” should contain a brief description of methods that were used for data analysis. This section must be improved. This part must describe the method of your work step by step and not describe what you can do with research - of course, this may remain in this part, because it enriches the description, but the key is to describe your work step by step.
  6. Findings - The results have been presented in a modest way. This would improve the results interpretation and explanation.
  7. Discussion and conclusion - To increase the significance of the results, the discussion part should embrace the differences and similarities among your findings and those of other scholars. The conclusion section should be a brief summary of article’s aim, methods and findings.

Detailed comments:

[33-35] this is true, but only in well-functioning organizations

[40- ...] why you write about fashion - if this is an example, I understand it, but where it is written, that it is an example

[44-47] Companies conduct business for profit not for the development of humanity.

[57] every abbreviation should be developed - even ICT

[62-68] this paragraph is unrelated to others

[71-75] it has no relation to the rest

[75] this is not an implication - the statement "hence" suggests a causal relationship

[77] these are goals - which are not discussed later - they are not elaborated, discussed and presented. It must be clearly marked, you write to achieve the goals.

[89-91] What does "previous research" mean?

- A lot of content appears in many places - surplus. Example: [256-266] this part of text is not related to your goals.

You can do that your text will contain a description of each article, but your goal is to systematize, arrange, present the main thoughts and ideas, show differences, present different, marginal (unclassified) directions of analysis and, as a consequence, show new directions of analysis.

Your article should be improved especially by achieving the objectives of the analysis, imposing structure on it and slightly reducing the content.

I really like your article and appreciate your work. It is interesting topic and the conclusions open the way for further research.

Good luck!

Author Response

Dear reviewer,

Thank you for your kind comments. We have revised our manuscript according to your comments. Please kindly find our detailer answers in the attachment and manuscript.

Thank you for reading.

Best wishes,

Fan Zeng
